# Health outcomes associated with Zika virus infection in humans: a systematic review of systematic reviews

Raphael Ximenes [ORCID],[1] Lauren C Ramsay,[1,2] Rafael Neves Miranda,[1] Shaun K Morris,[3,4] Kellie Murphy,[5] Beate Sander,[1,2,6,7] RADAM-LAC Research Team

**Correspondence to**
Dr Raphael Ximenes;
raphael.ximenes@theta.utoronto.ca

## ABSTRACT

**Objective** With the emergence of Zika virus (ZIKV) disease in Central and South America in the mid-2010s and recognition of the teratogenic effects of congenital exposure to ZIKV, there has been a substantial increase in new research published on ZIKV. Our objective is to synthesise the literature on health outcomes associated with ZIKV infection in humans.

**Methods** We conducted a systematic review (SR) of SRs following Preferred Reporting Items for Systematic Reviews and Meta-Analyses guidelines. We searched MEDLINE, Embase, Cochrane and LILACS (Literatura Latino-Americana e do Caribe em Ciências da Saúde) databases from inception to 22 July 2019, and included SRs that reported ZIKV-associated health outcomes. Three independent reviewers selected eligible studies, extracted data and assessed the quality of included SRs using the AMSTAR 2 (A MeaSurement Tool to Assess Systematic Reviews 2) tool. Conflicts were resolved by consensus or consultation with a third reviewer.

**Results** The search yielded 1382 unique articles, of which 21 SRs met our inclusion criteria. The 21 SRs ranged from descriptive to quantitative data synthesis, including four meta-analyses. The most commonly reported ZIKV-associated manifestations and health outcomes were microcephaly, congenital abnormalities, brain abnormalities, neonatal death and Guillain-Barré syndrome. The included reviews were highly heterogeneous. The overall quality of the SRs was critically low with all studies having more than one critical weakness.

**Conclusion** The evolving nature of the literature on ZIKV-associated health outcomes, together with the critically low quality of existing SRs, demonstrates the need for high-quality SRs to guide patient care and inform policy decision making.

**PROSPERO registration number** CRD42018091087.

## Strengths and limitations of this study

► Lack of systematic reviews (SRs) on Zika virus (ZIKV) in the literature.
► Lack of information about the risks of severe outcomes related to ZIKV infection or the presence of specific outcomes.
► Broad search strategy.
► Without restrictions by language or publication type.
► To our knowledge, this is the first SR of SRs about health outcomes associated with ZIKV infection in humans.

## INTRODUCTION

Zika virus (ZIKV) was first discovered in 1947 in rhesus monkeys in Uganda.[1] It is an arbovirus in the flavivirus family and typically causes mild illness in humans characterised by fever and rash. There were reports of sporadic cases of ZIKV infection in humans over the years in Asia and Africa,[2] with the first large documented outbreak taking place in Yap, a Micronesian Island, in 2007.[3] Since then, there have been reported outbreaks in French Polynesia (in 2013–2014), and most recently in South and Central America and the Caribbean.[4] With the emergence of ZIKV in Brazil, there were over 800 000 estimated cases of ZIKV infection reported by countries and territories in the Americas by January 2018.[5] By March 2017, according to the latest WHO global situation report on Zika, 84 countries, territories or subnational areas had evidence of vector-borne ZIKV transmission.[6] According to the Centers for Disease Control and Prevention (CDC), until May 2019, there were 89 areas with current or past transmission, but no current outbreak of ZIKV.[7]

Our understanding of Zika-associated clinical outcomes has evolved over time. Before human pathogenesis was understood, cellular level damage was apparent in animal studies in the 1950s.[8] The first study in humans to suggest an association between ZIKV and human disease was a case–control study during an outbreak in French Polynesia between 2013 and 2014, suggesting an association with Guillain-Barré syndrome (GBS).[9] However, the link between ZIKV in pregnant women and microcephaly in infants was only evident in the 2015–2016 outbreak in South America.[10] With the spread of ZIKV to new regions of the world and the extent of the

outbreak in South and Central American and Caribbean countries, a substantial body of new research has been published in recent years about Zika.

A bibliometric analysis of ZIKV research that indexed in Web of Science found a significant increase in the number of studies being published beginning in 2015 (n=38 publications) to 2017 (n=1962 publications).[11] Summarising the large body of literature on outcomes associated with ZIKV infection is timely and needed.

The purpose of this systematic review (SR) of SRs was to synthesise the currently known health outcomes associated with ZIKV infection in humans.

## METHODS

### Search strategy and selection criteria

We searched MEDLINE, Embase, Cochrane and LILACS databases from inception to 22 July 2019. Our search strategy across all databases included concepts related to 'Zika' and 'systematic review' (complete search strategy found in online supplementary file 1). Our search strategy was not restricted by language or publication type. Three reviewers (RX, first reviewer; LR and RM second reviewers) independently screened titles, abstracts and relevant full text of identified articles.

The inclusion criteria were defined as SRs that reported health outcomes of ZIKV infection in humans, that is, clinical presentation and sequelae of ZIKV infection in humans. We excluded studies that only reported symptoms (eg, rash and fever) of ZIKV infection, diagnostic techniques, mosquito control, therapeutic regimes, vaccine and trial but not outcomes (eg, GBS, congenital Zika syndrome). We followed the Preferred Reporting Items for Systematic Reviews and Meta-Analyses guidelines for reporting results.[12]

The data extraction was performed in duplicate by the reviewers. The SR methods were established prior to the conduct of the SR and there were no deviations from the protocol, except for adding the LILACS database to the search.

### Patient and public involvement

No patient involved.

### Quality appraisal

We used the AMSTAR 2 (A MeaSurement Tool to Assess Systematic Reviews 2) tool to critically appraise the included SRs.[13] AMSTAR 2 is not intended to generate an overall score, but rather to assist in the identification of high-quality SRs. Three reviewers (RX, first reviewer; LCR and RNM, second reviewers) independently evaluated the quality of each study based on weaknesses in critical domains as defined by the AMSTAR 2 tool. Studies were rated based on the overall confidence in the results of the SR and defined as either high (zero or one non-critical weakness), moderate (more than one non-critical weakness), low (one critical flaw with or without non-critical weaknesses) or critically low (more than one critical

flaw with or without non-critical weaknesses).[14] Critical domains included protocol registration, adequacy of the literature search, justification for excluding studies, risk of bias from individual studies included in the SR, appropriateness of meta-analytical methods, consideration of risk of bias when interpreting results and assessment of publication bias.[14] Any disagreements between the two reviewers were resolved by consensus.

### Data analysis

Three reviewers (RX, first reviewer; LCR and RNM, second reviewers) extracted the data using a structured electronic data extraction form, extracting study characteristics and measures of effect for outcomes resulting from ZIKV infection. Included studies were summarised narratively and health outcomes were reported where possible.

## RESULTS

We identified 1382 unique articles from the database searches (figure 1). After screening titles and abstracts, we selected 85 for full-text screening. Of these, 21 met our inclusion criteria.[15–35] The main reasons for exclusion at the full-text stage were articles not being SRs (but rather overviews or literature reviews) and studies only reported symptoms but not outcomes.

Study characteristics are summarised in table 1. The included SRs were published between February 2016 and May 2019. The types of studies eligible for inclusion in the SRs varied across studies; four SR did not include any information on the included studies,[21 24 28 30] all other SRs included observational studies (one limited to only cohort studies[18]) and the majority (71%; n=15) included case reports and case series. Three SRs considered evidence from modelling studies, animal experiments and in vitro experiments.[15 33 35] Another did not limit to reports of primary data and included SRs, narrative reviews and news articles.[20]

The majority of studies included in the SRs were conducted in Brazil, the USA, French Polynesia and Colombia.

### Summary of included SRs and outcomes

Of the 21 included SRs, the most commonly reported outcome was microcephaly, reported in 14 SRs,[15–26 30 32] 12 SRs reported on GBS,[15 16 19 20 22 23 25 27 29–31 33] 11 SRs reported on malformations or congenital abnormalities,[18–20 22 26 30–34] 9 reported on brain,[15 17 21 24–26 28 30 32] 7 SRs reported on ocular disorders[15 18 21 24 25 30 32] and 6 SRs on termination of pregnancy, fetal death and perinatal death.[15 18–20 30 33] Three SRs or fewer reported on auditory disorder,[15 26 34] cardiovascular damage,[18 26 35] neurological complications,[16 25 33] intrauterine growth restrictions,[15 25] abnormal amniotic fluid,[15] epilepsy[21] and death due Zika infection.[16]

Seven SRs focused on pregnant women[17–20 24 26 28] and five SRs included the general population,[15 16 22 23 29] while

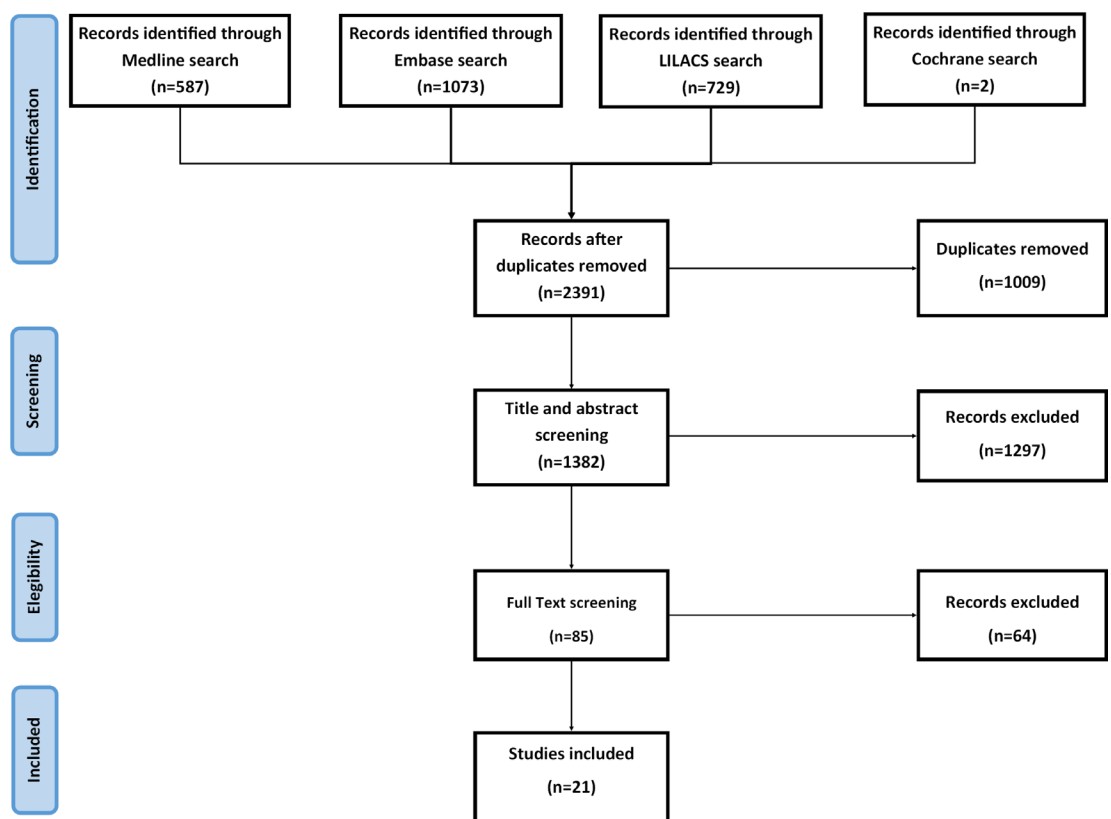

**Figure 1** Preferred Reporting Items for Systematic Reviews and Meta-Analyses flow diagram of search results and study selection.

newborns, neonates, perinatal, early birth or infants were included in five SRs.[18 19 21 25 26] One SR focused in travellers returning to the USA and Europe.[31] Adults were included in 2 of the 15 SRs.[25 27]

### Overlap between systematic reviews

Our SR includes 21 SRs. The overlap between the results of the 21 SRs included 860 studies (table 1), 615 of which were not duplicates. Out of the 615 studies, 477 (77.56%) were cited only once as studies included in the SRs included in our SR, and the remainder were cited in up to 10 SRs, 83 (13.50%) were cited twice, 29 (4.72%) three times, 8 (1.30%) four times, 8 (1.30%) five times, 6 (0.98%) six times, 2 (0.33%) seven times, 1 (0.16%) eight times and 1 (0.16%) ten times[3 36–52] (table 2, figure 2).

### Health outcomes

The online supplementary file 2 reports the health outcome data extracted from the 21 SRs.

#### Clinical outcomes associated with ZIKV infection during pregnancy

The online supplementary file 2 shows that the reported outcomes associated with ZIKV infection during pregnancy ranging from adverse birth outcomes to perinatal death. The frequency of infant deaths (miscarriages, perinatal deaths, intrauterine death or stillbirth and termination of pregnancy) was reported by 6 of 21 SRs,[15 18–20 30 33] ranging from 4.8% to 22%.

Congenital Zika syndrome (CZS) was reported in many different ways. Some studies reported specific outcomes related to CZS (eg, brain abnormalities, ocular disorder or microcephaly) while others reported CZS as a nonspecific outcome. The prevalence of CZS ranged from 2% (5 cases in 250 ZIKV-infected pregnant women)[18] to 50% (58 adverse congenital outcomes out of 117 women with PCR confirmed ZIKV).[22]

Brain abnormalities were explicitly reported with data from 19 studies in which 96% (205 in 213 pregnant women) of fetuses were diagnosed after confirmation with imaging tests.[15] One SR reported the prevalence of brain abnormalities (28%) including microcephaly in newborns whose mothers were infected with ZIKV in pregnancy[25] while other SR reported an observational study of 35 infants with microcephaly, 11 fetuses had intrauterine brain injury accompanied by stunting of cerebral growth prior to birth.[17] Further, five SRs classified the type of brain abnormalities or where the lesions were found[21 24 28 30 32] as intracranial calcification, reduction in the constitution of gyri of the severe cerebral cortex, abnormal hypodensity of the white matter, malformations of cortical development, subcortical–cortical junction calcifications, basal ganglia calcification, brain calcification, intraventicular synechiae and periventricular cystic, brain volume loss, ventriculomegaly/hydrocephaly and diffuse involvement of all the cerebral lobes.

Microcephaly was reported in 14 of 21 SRs. Chibueze et al[17] provided a trimester-specific modelling estimate risk for microcephaly. When the infection occurs in an indeterminate period of pregnancy, ZIKV-associated microcephaly

**Table 1** Summary of included SRs

| Author (year) | Aim | Search period | Number of studies included | Types of studies included in review | Jurisdictions of included studies (n studies) |
|---|---|---|---|---|---|
| Krauer et al (2017)[15] | To assess the relationship between ZIKV infection and congenital brain abnormalities and Guillain-Barré syndrome | From inception until 30 May 2016 | 106 | Case reports, case series, case–control studies, cohort studies, cross-sectional studies, ecological study/outbreak reports, modelling studies, animal experiments, in vitro experiments, sequence analysis and phylogenetics | Brazil (6), Cabo Verde (2), Colombia (1), French Polynesia (2), Martinique (2), Panama (5), El Salvador (1), Haiti (119), Puerto Rico (1), Venezuela (1), Slovenia*, Netherlands*, Dominican Republic*, French Guiana*, Honduras*, Paraguay*, Suriname*, Micronesia*, Pacific Islands* |
| Paixão et al (2016)[16] | To summarise current knowledge on ZIKV including epidemiology, clinical presentation and complications | 1954 to January 2016 | 41 | Case reports, case series, surveillance reports, cross-sectional studies, epidemiological bulletins and alerts | Not clearly reported Most data are from Brazil and French Polynesia |
| Chibueze et al (2017)[17] | To summarise guidance on pregnancy care in the context of ZIKV infection | From inception until 3 March 2016 | 18 | Case reports, case series, observational studies | Brazil (11) Colombia (1) France (1) Puerto Rico (1) Slovenia (1) USA (2) Venezuela (1) |
| Coelho et al (2017)[18] | To summarise evidence and meta-analyse data to estimate prevalence of microcephaly in babies born to ZIKV-infected pregnant women | Not reported | 8 | Cohort studies | Brazil (1) Colombia (1) French Guiana (1) Puerto Rico (1) USA (4) |
| Simões et al (2016)[19] | To assess the effects of ZIKV infection on during pregnancy and postpartum periods | From inception until 23 February 2016 | 30 | Case reports, case series, guidelines | Not clearly reported; most data are from Brazil |
| Padilla et al (2016)[20] | To review clinical and basic science literature about ZIKV infection relevant for obstetric anesthesiologists | From inception until 15 April 2016 | 30 | SRs, narrative reviews, case reports, epidemiological studies, government reports and news articles | Not clearly reported |
| Marques et al (2019)[21] | To map the neurological damage and outcomes related to congenital ZIKV infection | January 1966 to August 2018 | 28 | Not informed | Brazil (16) USA (3) Colombia (1) |
| Counotte et al (2018)[22] | To summarise the evidence of the casual associations between ZIKV and CZS and GBS | 30 May 2016 to 18 January 2017 | 101 | Case report, case series, case–control study, cohort study, cross-sectional study, controlled trials, ecological study/outbreak report, modelling study, animal experiment, in vitro experiment, sequencing and phylogenetics, biochemical/protein structure studies | USA, Martinique, Brazil, Suriname, Colombia, French Guiana, Slovenia, Spain, Uganda, Nicaragua, Barbados, Belize, Dominican Republic, El Salvador, Guatemala, Haiti, Honduras; Mexico, Republic of Marshall Islands, Venezuela, French Polynesia, Ecuador, France, Puerto Rico, Guadeloupe, Guyana, New Zealand, French Southern Territories. |
| Haby et al (2018)[23] | To estimate and meta-analyse the prevalence of asymptomatic ZIKV infection in the general population and in specific population groups from observational epidemiological studies | From inception until 26 January 2018 | 23 | Cross-sectional seroprevalence studies, case series, case–control, cohort | USA (6), Brazil (3), French Polynesia (3), French Guiana (3), Puerto Rico (2), Colombia (2), Spain (2), Micronesia (1), Martinique (1) |
| Sarwar et al (2018)[24] | To report on the current literature regarding ZIKV and its hazardous effects on maternofetal health with a special emphasis on risk assessment, virus transmission, associated complications and possible management | 2007 to May 2017 | 69 | Not informed | Argentina, Bolivia, Brazil, Colombia, French Guiana, Suriname, Paraguay, Trinidad and Tobago, Canada, Dominican Republic, Grenada, Guadeloupe, Guatemala, Haiti, Martinique, Puerto Rico, USA, Costa Rica, El Salvador, Honduras, Nicaragua, Panama, Europe, Slovenia, Spain, Thailand, Vietnam, French Polynesia, Marshall Islands, Cape Verde. |
| Wahid et al (2018)[25] | To summarise the evidence of neurological complications in ZIKV-infected people | 2015 to March 2017 | 35 | Case-studies, case-cohort studies, cross-sectional studies, organisational survey reports and case–control studies | Brazil (15) French Polynesia (3) Colombia (3) USA, Slovenia, Suriname, Spain, Haiti, Martinique, Netherlands, Ecuador, Guyana (1) |

Continued

**Table 1** Continued

| Author (year) | Aim | Search period | Number of studies included | Types of studies included in review | Jurisdictions of included studies (n studies) |
|---|---|---|---|---|---|
| Soriano-Arandes et al (2018)[26] | To summarise the new evidence in aspects of epidemiology, virology, pathogenesis, associated risk factors during pregnancy, newborn phenotypic signs, neuroimaging, laboratory diagnosis, treatment and vaccines | From inception until 30 November 2017 | 106 | Case series, cohort (prospective/retrospective), cross-sectional or case-control studies | Brazil, French Polynesia, USA, Martinique, Colombia |
| Barbi et al (2018)[27] | To systematically review the literature and perform a meta-analysis to estimate the prevalence of GBS among ZIKV-infected individuals | From inception until November 2017 | 3 | Case series, epidemiological surveys, cross-sectional or cohort studies | French Polynesia (1), Suriname and Dominican Republic (1), South American, Central American and Caribbean countries (1) |
| Santos et al (2018)[28] | To analyse the association between Zika-virus and microcephaly during the gestational period | From inception until December 2016 | 35 | Not informed | Brazil |
| Wachira et al (2018)[29] | To describe the factors associated with development of GBS, both infectious and non-infectious, through an SR | 1 January 2007 to 30 June 2017 | 34 | The most common were case control, cohort and self-controlled case series | French Polynesia |
| Pomar et al (2019)[30] | Present a review to describe the risks and complications of maternal and subsequent fetal infection by ZIKV | June 2009 to November 2018 | 68 | Not informed | Colombia (3), Puerto Rico (1), French Guiana (3), Brazil (1), Yap Island (1), USA (2) |
| Wilder-Smith et al (2018)[31] | Describe the burden of ZIKV infection in international travellers over time; estimate the proportion of birth defects as a result of maternal ZIKV infection in travellers; track the extent of sexual transmission; summarise data on ZIKV cases in travellers identifying counties with reports on local transmission | 1947 to April 2017 | 65 | Surveillance reports, case reports, retrospective (multicentre study), descriptive retrospective analysis and prospective cohort study | USA (9), Canada (2), Germany (3), Norway (1), France (5), Italy (7), Japan (2), Australia (4), New Caledonia (1), Finland (1), Mexico (1), Slovenia (1), Netherlands (4), Belgium (1), Portugal (1), Switzerland (3), Israel (1), Taiwan (2), Spain (1), China (7), South Korea (2), UK (2), Singapore (1), Malaysia (1)USA (9), Canada (2), Germany (3), Norway (1), France (5), Italy (7), Japan (2), Australia (4), New Caledonia (1), Finland (1), Mexico (1), Slovenia (1), Netherlands (4), Belgium (1), Portugal (1), Switzerland (3), Israel (1), Taiwan (2), Spain (1), China (7), South Korea (2), UK (2), Singapore (1), Malaysia (1)USA (9), Canada (2), Germany (3), Norway (1), France (5), Italy (7), Japan (2), Australia (4), New Caledonia (1), Finland (1), Mexico (1), Slovenia (1), Netherlands (4), Belgium (1), Portugal (1), Switzerland (3), Israel (1), Taiwan (2), Spain (1), China (7), South Korea (2), UK (2), Singapore (1), Malaysia (1) |
| Nithiyanantham et al (2019)[32] | To conduct an SR and meta-analysis on the prevalence of congenital Zika-related disorders in infants of mothers infected with ZIKV during pregnancy | From inception until 31 October 2017 | 25 | Case series, epidemiological reports, prospective and retrospective studies, cohort studies and cross-sectional studies | USA (8), Brazil (6), Colombia (2), Puerto Rico (1), French Polynesia (1), Martinique (1), Trinidad and Tobago (1), French Guiana (1), Ecuador (1), Spain (1) |
| Masel et al (2019)[33] | To determine if prior infection with DENV, as compared with those with no prior DENV infection, is associated with a greater risk of ZIKV complications (including neurological and teratogenic outcomes), greater ZIKV peak viremia, greater area-under-the-curve of viremia or other putative laboratory proxies of ZIKV severity | From inception until 25 March 2018 | 5 | Case–control study | Brazil (2), French Polynesia (5) |
| Barbosa et al (2019)[34] | To describe the auditory alterations, pathogenesis and recommendations for follow-up in individuals with prenatal or acquired ZIKV infection | From inception until April 2019 | 27 | Case report and case series | Brazil (14), Colombia (3), USA (2), French Polynesia (1), Puerto Rico (1) |
| Minhas et al (2017)[35] | Focuses on the potential threat that ZIKV may pose to the heart like that of similar arboviral diseases | From inception until March 2017 | 3 | Case report and prospective observational multicentre study | France (1), Venezuela (1), China (1) |

*Not possible to know number of studies from these countries.
GBS, Guillain-Barré syndrome; SR, systematic review; ZIKV, Zika virus.

**Table 2** Overlap between systematic reviews

| Number of citations | Title | Author (year) | Cited by |
|---|---|---|---|
| 10 | ZIKV infection in pregnant women in Rio de Janeiro | Brasil et al (2016)[36] | 15 17 18 20–22 25 26 30 32 |
| 8 | ZIKV associated with microcephaly | Mlakar et al (2016)[37] | 15 17 19 24–26 28 31 |
| 7 | Ocular findings in infants with microcephaly associated with presumed ZIKV congenital infection in Salvador, Brazil | de Paula Freitas et al (2016)[38] | 15 17 19–21 25 30 |
| 7 | Possible association between ZIKV infection and microcephaly—Brazil, 2015 | Schuler-Faccini et al (2016)[39] | 15 17 21 25 26 28 30 |
| 6 | Guillain-Barré syndrome outbreak associated with ZIKV infection in French Polynesia: a case–control study | Cao-Lormeau et al (2016)[40] | 15 20 23–25 33 |
| 6 | Birth defects among fetuses and infants of US women with evidence of possible ZIKV infection during pregnancy | Honein et al (2017)[41] | 18 21 22 24 26 32 |
| 6 | ZIKV infection among US pregnant travellers—August 2015—February 2016 | Meaney-Delman et al (2016)[42] | 15 17 18 20 31 32 |
| 6 | ZIKV outbreak on Yap Island, Federated States of Micronesia | Duffy et al (2009)[3] | 15 16 19 23 24 30 |
| 6 | Association between ZIKV and microcephaly in French Polynesia, 2013—15: a retrospective study | Cauchemez et al (2016)[43] | 15 17 24–26 30 |
| 6 | ZIKV intrauterine infection causes fetal brain abnormality and microcephaly: tip of the iceberg? | Oliveira et al (2016)[44] | 15 17 20 21 26 28 |
| 5 | Congenital cerebral malformations and dysfunction in fetuses and newborns following the 2013 to 2014 ZIKV epidemic in French Polynesia | Besnard et al (2016)[45] | 15 25 30 32 34 |
| 5 | Description of 13 infants born during October 2015–January 2016 with congenital ZIKV infection without microcephaly at birth—Brazil | van der Linden et al (2016)[46] | 21 22 26 30 34 |
| 5 | Congenital brain abnormalities and ZIKV: what the radiologist can expect to see prenatally and postnatally | Oliveira-Szejnfeld et al (2016)[47] | 21 22 26 30 32 |
| 5 | Detection and sequencing of ZIKV from amniotic fluid of fetuses with microcephaly in Brazil: a case study | Calvet et al (2016)[48] | 15 17 19 28 30 |
| 5 | Congenital ZIKV syndrome in Brazil: a case series of the first 1501 livebirths with complete investigation | França et al (2016)[49] | 21 22 25 26 30 |
| 5 | Evidence of perinatal transmission of ZIKV, French Polynesia, December 2013 and February 2014 | Besnard et al (2014)[50] | 16 17 24 26 28 |
| 5 | Increase in reported prevalence of microcephaly in infants born to women living in areas with confirmed ZIKV transmission during the first trimester of pregnancy—Brazil, 2015 | Oliveira et al (2016)[51] | 15 17 20 24 25 |
| 5 | ZIKV infection complicated by Guillain-Barré syndrome—case report, French Polynesia, December 2013 | Oehler et al (2014)[52] | 15 16 19 20 25 |

ZIKV, Zika virus.

was described by Coelho et al[18] The authors performed a meta-analysis and found a prevalence of 2.3% (95% CI 1% to 5.3%) of microcephaly when considering all pregnancies (2941 mother–infant pairs). When considering only live births (2648 live births), the prevalence of microcephaly was 2.7% (95% CI 1.2% to 6%).[18] Nithiyanantham et al[32] also performed a meta-analysis of the proportion of congenital disorders in infants born to ZIKV-infected mothers, reporting a prevalence of 3.9% (95% CI 2.4% to 5.4%).[32] Pomar et al[30] reported the prevalence of microcephaly in CZS ranging from 33.3% to 64%.[30] Four SRs reported microcephaly cases per livebirth pregnancies, ranging from 0.2% (20 cases per 10 000 live births) to 14.3% (1 case in seven livebirth pregnancies)[15 16 18 20] and one SRs reported

10 microcephaly cases per 10 000 births.[19] Microcephaly risk in infected pregnant women was reported in four SRs. The absolute risk varied between 0.95% (95% CI 0.34% to 1.91%) during the first trimester of pregnancy to 30%[22 24–26] (trimester not reported). Death caused by microcephaly was estimated in a study reported by Coelho et al[18] reporting a rate of 8.3% (171 deaths among 2063 confirmed cases of microcephaly).[18] The prevalence of microcephaly in asymptomatic ZIKV infection was also reported as 0.36% (0.22%–0.51%).[23] Another SR reported that in a series of 13 infants with congenital ZIKV infection and microcephaly, more than half of the mothers did not report any symptoms of ZIKV prior to delivery.[26]

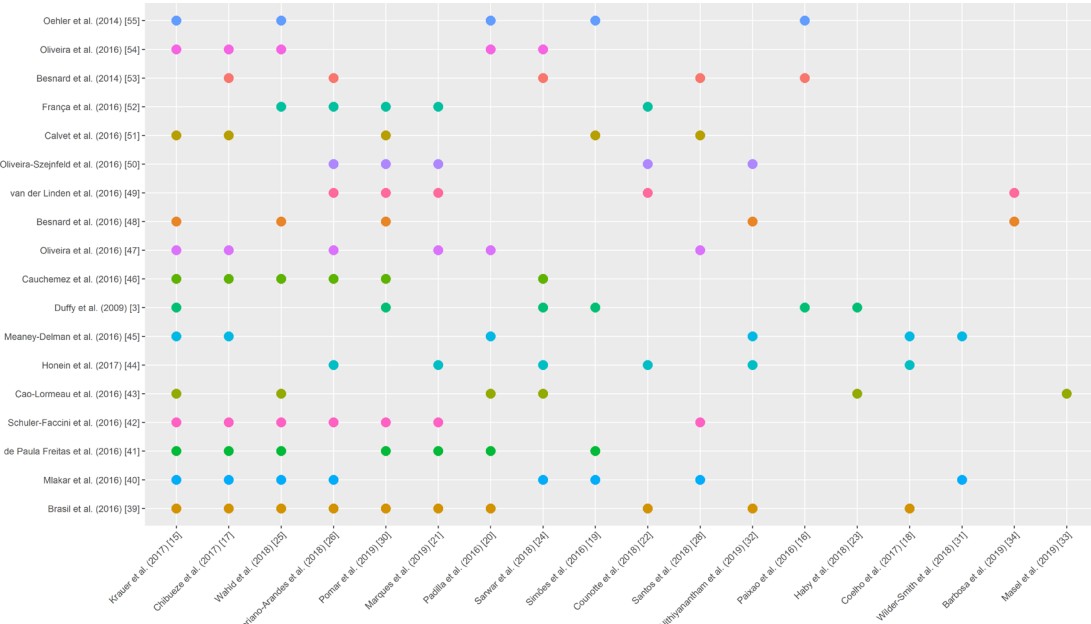

**Figure 2** Overlap between studies cited in at least five systematic reviews.

The prevalence of congenital ZIKV syndrome-related outcomes is still unknown. In this SR of SRs, we found the intrauterine growth restrictions rate reported varied from 28.57% (10 cases in 35 mother–infant pairs)[15] to 31.43%, from one observational study of 35 infants with microcephaly.[17] Another study reported intrauterine growth restriction in 11.9% of fetuses with or without microcephaly (5 fetuses from 42 positives for ZIKV pregnant women).[25] Pomar et al[30] reported the prevalence of intrauterine growth restriction in 14% of CZS cases. The prevalence of ocular disorder was reported in five SRs ranging from 0.9%% (from one study with 395 livebirth pregnancies) to 58.6% (17 ocular findings with microcephaly associated in 29 infants).[15 18 21 24 25 30 32] Abnormal amniotic fluid was described only by Krauer et al.[15] Auditory disorder was described by Krauer et al[15] (prevalence of 13%—3 cases in 24 mother–infant pairs) and Soriano-Arandes et al[26] (prevalence of 7%—5 cases in 70 children with laboratory diagnosis of ZIKV infection) and Barbosa et al[34] (variations in the frequency of altered otoacoustic emissions testing (OAE) and automated auditory brainstem (a-ABR) response testing across the studies in 515 children: altered OAE varied from 0% to 75%, while altered a-ABR varied from 0% to 29.2%). The prevalence of cardiovascular damage was reported by Coelho et al[18] (prevalence of 1%—3 cases in 301 livebirth pregnancies), Soriano-Arandes et al[26] (prevalence of 13.6%—14 cases in103 ZIKV cases) and Minhas et al[35] (prevalence of 67% of heart failure in a cohort with nine adults positive for ZIKV and no previous cardiac history).

### Neurological complications associated with ZIKV infection

Neurological complications were reported by 12 of 21 SRs,[16 19–23 25 27 29–31 33] where GBS was the most commonly reported neurological complication.

Among adults, the proportion of neurological complications associated with ZIKV infection in Bahia (Brazil) was similar to that in French Polynesia. Among these neurological complications, GBS was diagnosed in 1 of every 1000 reported Zika cases in Brazil and 1.3 per 1000 in French Polynesia.[16] During the French Polynesia outbreak in 2013, the incidence of GBS has been 0.24 per 1000 ZIKV infections,[20] and Simões et al[19] described one case report in French Polynesia in which GBS was diagnosed in a patient with ZIKV.[19]

Counotte et al[22] reported the increased incidence of GBS incidence ratio between during and pre-ZIKV outbreak periods in seven different countries; which ranged from 2.0 (95% CI 1.6 to 2.6) to 9.8 (95% CI 7.6 to 12.5), while Barbi et al[27] conducted a meta-analysis of the prevalence of GBS in ZIKV-infected cases. Their estimate for the prevalence of GBS in adults infected with ZIKV was 1.23% (95% CI 1.17%–1.29%). This same study was reported by Pomar et al[30]. Krauer et al[15] reported the prevalence of symptomatic ZIKV in GBS cases (74%–84% symptomatic ZIKV in GBS cases). Paixão et al,[16] Padilla et al[20] and Barbi et al,[27] described the prevalence of admission to an intensive care unit (ranging from 36% to 42%, among 42 and 38 GBS cases, respectively) and mechanical ventilation (21% to 29% among 42 GBS cases) in French Polynesia. The interval between ZIKV and GBS symptoms was described by Krauer et al,[15] Paixão et al,[16] Padilla et al[20] and Counotte et al.[22] The highest interval was reported by Paixão et al[16] where 88% of GBS cases reported a viral syndrome up to 23 days before the onset of the neurological syndrome. No deaths due to GBS related with ZIKV infections were reported in this SR.

Epilepsy and sleep profiles were described in two SRs. For Marques et al (2019), the prevalence of epilepsy in congenital ZIKV infants ranged from 42% (43 in

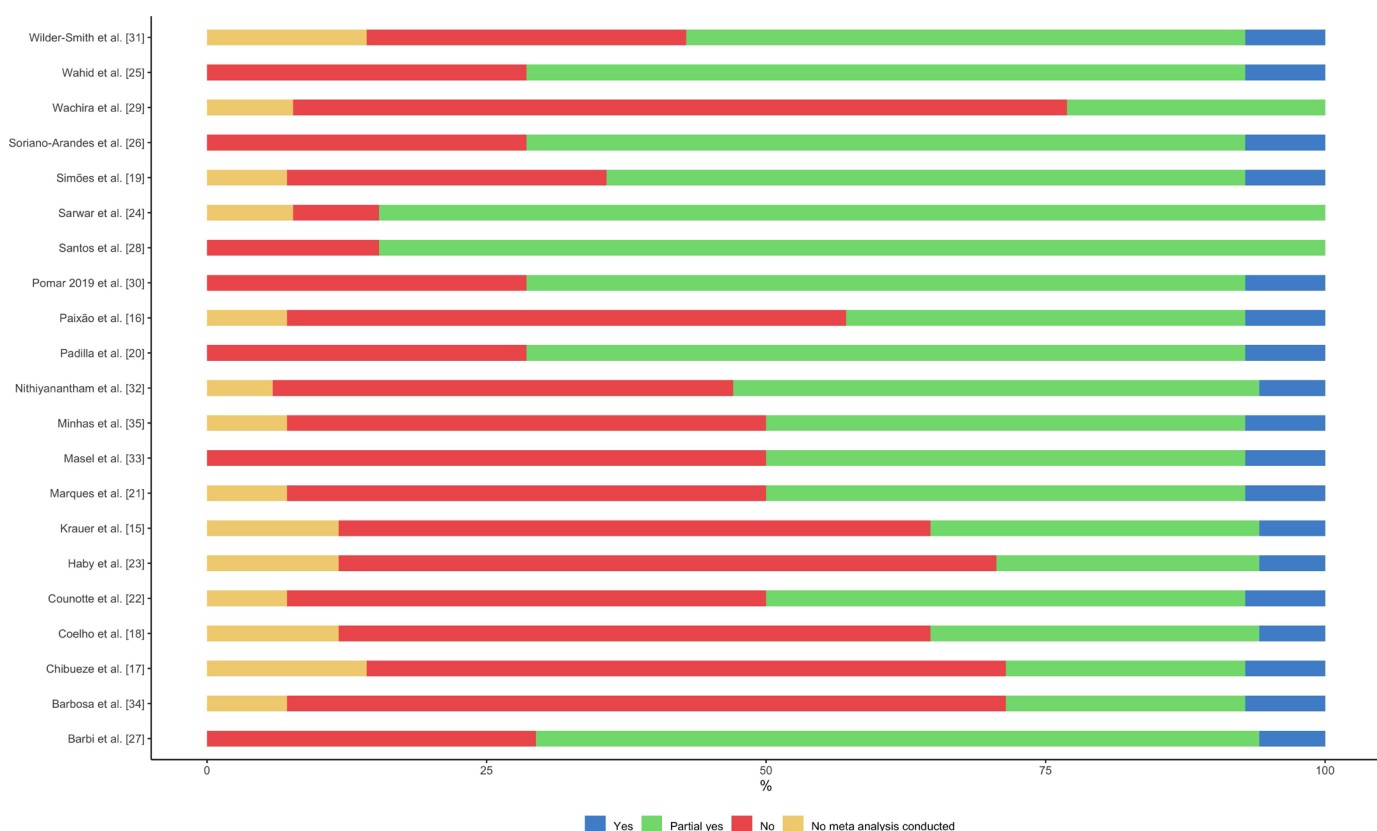

**Figure 3** Individual study results of quality assessment using AMSTAR 2 (A MeaSurement Tool to Assess Systematic Reviews 2)—result for all questions of AMSTAR 2 tool.

102 children with congenital ZIKV) to 67% (95 in 141 congenital ZIKV) and 34% (30 in 88 congenital ZIKV-infected children) of the ZIKV-infected children were defined as poor sleepers.[21] Pomar *et al*[30] reported that 9% to 95.5% of congenital ZIKV infections were associated with epilepsy.

Idiopatic thrombocytopaenia purpura (ITP) related with ZIKV infection was reported by Counotte *et al.*[22] They reported 11 cases of ITP across 18 studies; however, there is no information about the total number of ZIKV-infected subjects in these studies.

### Deaths associated with ZIKV infection

Deaths due to Zika infection are rare. According to the Brazilian Ministry of Health, between 440 000 and 1 300 000 cases of Zika occurred in Brazil in 2015.[53 54] Since the beginning of the outbreak 11 deaths among adults were confirmed in Brazil and an additional nine deaths were reported by the countries and territories in the Americas.[5]

### Coinfection

Coinfection was reported with dengue,[16–18 25] chikungunya[16 17 25] and HIV[16 17]; cytomegalovirus (CMV), toxoplasmosis or other known teratogenic agents[16–18]; hepatitis B virus, hepatitis C virus, CMV, herpes simplex virus, Epstein-Barr virus, rubella, human T lymphotrophic virus, parvovirus B19 and syphilis.[17]

Masel *et al*[33] found no association of prior exposure to Dengue Virus (DENV) and fetal loss, or clinical neurological assessment of fetus, and no statistical difference in prior DENV-exposed patients with or without GBS after ZIKV infection.

### Quality assessment

Of the 21 SRs included, there was high inter-rater reliability between the reviewers (91%). The overall quality of the SRs was critically low with all studies identified as having more than one critical weakness with or without non-critical weaknesses (figure 3). For all studies, the majority (65%) of answers for the six critical domains of AMSTAR 2 tool (questions 2, 4, 7, 9, 12 and 14) were 'no' or 'partial yes' (53% and 12%, respectively) (figure 4 and online supplementary file 3). Main weaknesses identified were a deficient bibliographic search strategy and the lack of an explicit statement that SR methods were established prior to the conduct of the SR.

### DISCUSSION

Our SR of SRs identified 21 SRs that reported health outcomes associated with ZIKV infection. Microcephaly was the most commonly reported health outcome. Other outcomes reported were fetal death, neonatal death, congenital abnormalities including brain abnormalities, intrauterine growth restrictions, ocular disorders and

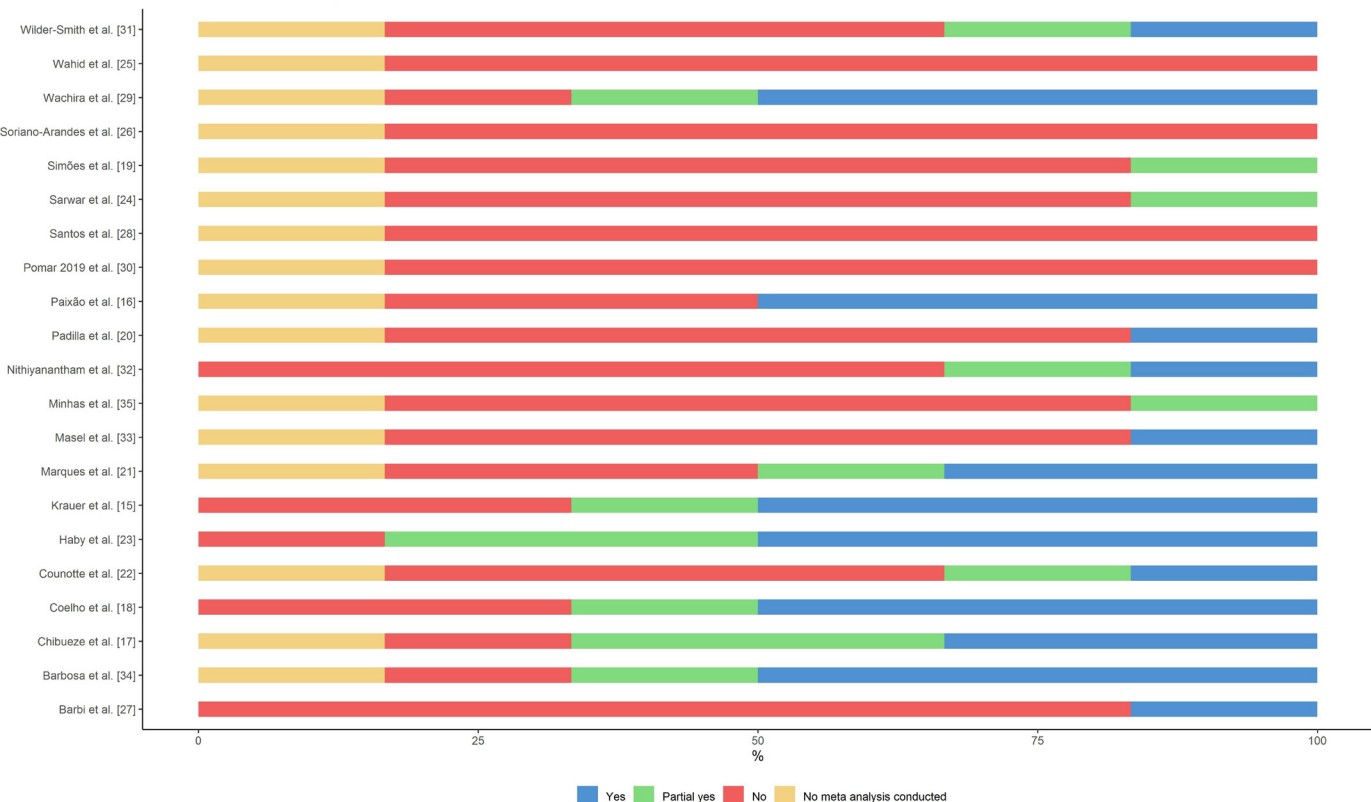

**Figure 4** Individual study results of quality assessment using AMSTAR 2 (A MeaSurement Tool to Assess Systematic Reviews 2)—critical domains of AMSTAR 2 tool.

infant disorders including auditory disorders, cardiovascular damage, death due ZIKV infection, neurological complications, epilepsy and finally adult outcomes including GBS. The included SRs indicate that ZIKV infection is causally associated with congenital abnormalities, including microcephaly, and that ZIKV infection is a trigger of GBS, considering evidence on biological plausibility, the strength of association and the exclusion of alternative explanations.

Overall, we found high heterogeneity among the 21 included SRs ranging from descriptive SRs, with few data on health outcomes associated with ZIKV infection, to more quantitative SRs, including four meta-analyses. There was some overlap (22%) of included studies across the SRs, indicating that the SRs are relatively distinct from each other and consistent with the included SRs reporting on different aspects of ZIKV infection. Given this heterogeneity, it was not possible to perform a quantitative synthesis, making it difficult to compare the results or draw conclusions based on the included SRs. Further, our quality appraisal found that all SRs were of critically low quality, with only three or fewer of six critical domains of AMSTAR 2 tool met in any study.

Further research into the magnitude of effects, potential other immediate and late outcomes, and long-term sequelae is warranted to understand the full impact of ZIKV infection, particularly long-term follow-up studies of infants born to ZIKV-infected mothers and infants and children infected with ZIKV early in life. In a recent study,

Nielsen-Saines et al[55] reinforce this conclusion. They observed that the neurological phenotype in some ZIKV-exposed children may change from abnormal to normal from birth into early childhood and vice versa.[55]

Our SR has some limitations. Since ZIKV is an emerging disease, and despite the increasing number of SRs, one limitation is the lack of SRs on ZIKV in the literature. Because the Brazilian outbreak prompted much of the recent research, 7 of 21 (33%) included SRs were conducted fairly early in the epidemic between 2016 and 2017, 43% in 2018 and 24% in 2019, which can explain the lack of information on severe outcomes related to ZIKV infection or the presence of specific outcomes, caused by the inability to observe outcomes that are only evident or possible to detect in older children. Often the reported data are unclear as to the nature of the infection, that is, whether included subjects are suspected ZIKV cases or confirmed ZIKV cases. Further, some of the included SRs did not report denominators, making interpretation difficult.

The low quality of the included SRs may indicate an important publication bias related to rare (eg, ITP) or poorly reported outcomes (eg, sleep disorders, epilepsy and auditory disorder) as these may not be captured in the search strategy.

Our study was strengthened by using a broad search strategy, without restrictions by language or publication type, reducing selection bias. To our knowledge, this is the first SR of SRs about health outcomes associated with ZIKV infection in humans.

As SRs of SRs aim to provide a summary of evidence from other SRs, although we were not able to perform a meta-analysis, our SR synthesises findings from SRs on health outcomes associated with ZIKV infection in humans.

The evolving nature of the literature on ZIKV-associated health outcomes together with the critically low quality of existing SRs, confirm the need for high-quality SRs to better understand the burden of ZIKV, guide patient care and inform health policy.

## CONCLUSION

Our SR demonstrates the need for future SRs on health outcomes associated with ZIKV infection as more research is published. As the ZIKV epidemic continues to evolve and the time since the emergence of the Brazilian outbreak increases we expect more primary observational studies on associated short- and long-term health outcomes to be published and synthesised in future SRs.

**Author affiliations**
[1]Toronto Health Economics and Technology Assessment (THETA) Collaborative, University Health Network, Toronto, Ontario, Canada
[2]Institute of Health Policy, Management and Evaluation, University of Toronto, Toronto, Ontario, Canada
[3]Division of Infectious Diseases and Centre for Global Child Health, Hospital for Sick Children, Toronto, Ontario, Canada
[4]Department of Pediatrics, Faculty of Medicine, University of Toronto, Toronto, Ontario, Canada
[5]Department of Obstetrics and Gynecology, Mount Sinai Hospital, Toronto, Ontario, Canada
[6]Public Health Ontario, Toronto, Ontario, Canada
[7]ICES, Toronto, Ontario, Canada

**Acknowledgements** The authors would like to acknowledge Andrea Tricco and Joanna Bielecki for their guidance on this project.

**Collaborators** RADAM-LAC Research Team members are Beate Sander, Camila Gonzalez, Manisha Kulkarni, Marcos Miretti, Mauricio Espinel, Jianhong Wu and Varsovia Cevallos.

**Contributors** RX: conceptualisation of the study, performed the systematic review, critically appraising the scientific literature, analysis, drafting and revising the manuscript. RNM: performed the systematic review, critically appraising the scientific literature and revising the manuscript. LCR: performed the systematic review and critically appraising the scientific literature. SKM: critical revision of the manuscript. KM: critical revision of the manuscript. RADAM-LAC Research Team: contribution to study conception and design. BS: conceptualisation of the study, critical revision of the manuscript and supervision of the study.

**Funding** This project was funded by the Canadian Institutes of Health Research—Team grant—FRN149784 and the International Development Research Centre (IDRC).

**Competing interests** None declared.

**Patient consent for publication** Not required.

**Provenance and peer review** Not commissioned; externally peer reviewed.

**Data availability statement** All data relevant to the study are included in the article or uploaded as supplementary information.

**ORCID iD**
Raphael Ximenes http://orcid.org/0000-0003-2536-951X

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
