## [Reviewer comments · BMJ Open]

ARTICLE DETAILS

TITLE (PROVISIONAL)	Health outcomes associated with Zika virus infection in humans: a systematic review of systematic reviews
AUTHORS	Ximenes, Raphael; Ramsay, Lauren; Miranda, Rafael; Morris, Shaun; Murphy, Kellie; Sander, Beate

VERSION 1 – REVIEW

REVIEWER	Lavinia Schuler-Faccini Universiade Federal do Rio Grande do Sul
REVIEW RETURNED	02-Jul-2019

GENERAL COMMENTS	This is an interesting manuscript that summarizes correctly the outcomes related to ZIKV infection, mostly after gestational exposures. Moreover, performing a systematic review of systematic reviews the authors were able to detect the weaknesses in other published papers with similar approach. Another strength is that they didn't had limits to English language; in this particular topic it's very helpful since there are publications in Portuguese and Spanish. I have only one minor observation: p3; In 26-27 (ref 10) the authors mention that the ZIKV and associated microcephaly occurred in 2016 in South America. Actually it was 2015-2016. In Brazil the peak of microcephaly was in 2015 decreasing in 2016. Colombia and other countries followed the peak in 2016. Although the formal establishment of ZIKV as a potential teratogen was in 2016, it was largely based from observations of babies born in 2015.
---

REVIEWER	Michel Counotte Institute of Social and Preventive Medicine, University of Bern, Bern, Switzerland
REVIEW RETURNED	15-Jul-2019

GENERAL COMMENTS	Ximenes et al. describe the health outcomes associated with Zika virus (ZIKV) infection in humans. They use a systematic review of systematic reviews (SR) to make a systematic inventory of the available published work indexed in the MEDLINE, Embase or Cochrane database up to February 13, 2019. They provide an overview of the different ZIKV-associated health outcomes from 15 SRs. The authors describe the results presented in individuals SRs. They conclude that higher quality SR are required, based on an evaluation of the quality using the AMSTAR2 tool. The authors take on the enormous task of navigating through 15 SR reporting on 667 studies (likely with quite some overlap), which should be applauded. However, the description of the studies and discussion as it is now should be improved.
---

	Introduction Page 3, line 38 on the purpose of the study: Do the authors mean a causal association between ZIKV and health outcomes? For the sake of public health guidance, the definition of the type of association might be relevant and at least worth to clarify. Methods We are currently almost half a year after the inclusion date, would the authors consider updating the review? Would the authors consider searching “LILACS” as an additional literature database? Results General comment: A more intuitive structure might be to present the results first and the risk of bias assessment last? One could argue that ‘risk of bias assessment’ is not the right way to describe the AMSTAR2 conclusions, since these partly check reporting quality, and partly bias. Overview of included studies: • It seems that the SRs included are conducted with different research questions or PICO’s guiding them. It would be great to provide an overview of these in the supplementary material. • Is there a reason for example Prata-Barbosa (2018) (http://www.scielo.br/pdf/jped/v95s1/0021-7557-jped-95-s1-0s30.pdf) was not included (review on the effect of Zika on growth)? • It seems that on page 4, line 52 the authors distinguish between reported symptoms and outcomes, and use this as exclusion criterium, but continue to discuss reported symptoms as a health outcome. This seems contradictory. Either revisit the inclusion criteria or omit discussing symptoms. Quality of included studies/AMSTAR rating • Since the authors worked with independent reviewers: how was the agreement and disagreement between the reviewers in AMSTAR2 rating? Does this provide reasons for discussion? Can the authors discuss the performance of the tool? • Can the authors provide insight in the AMSTAR2 rating, a more complete description, for example a supplementary table with ratings. The current figure only provides the percentage of scores within different categories. There is currently no way to see on which domains studies scored poorly or could be improved, this might also facilitate the discussion of the results, which is currently lacking. • It is unclear which items are the critical domains in figure 2. Please clarify. Page 5 line 23 only mentions ‘the majority’. Since one of the main conclusions is on the quality of studies, this needs to be expanded. A table in the main text might be appropriate. • Does this quality rating satisfy all the needs? Are the authors interested in the health outcomes related to suspected ZIKV or confirmed ZIKV? Is this worth discussing as limitation that we are often not sure on the infection status? Overview of Health outcomes General remarks: • It seems that the authors mainly describe how many of the included SRs describe a certain health outcome, would the authors agree that a systematic description of the actual incidence or prevalence would be more informative? • We numbers are provided, denominators are often unclear: the authors report a ‘prevalence’ of an outcome, but fail to mention the
--	---

denominator, making it difficult to interpret, consistency and clarification throughout the results would be in place.

- A structured description of the results per outcome in the form of a table might clarify the reported estimates, and heterogeneity. The text seems to be a somewhat random inclusion of numbers that are rather hard to follow.
- The definition of 'congenital abnormalities' is confusing and seems to include microcephaly in some cases, but not in others?

Specific remarks:

- Page 5, line 36-37 malformations or congenital abnormalities and brain abnormalities: the latter could be included in the first? Again: clarifying terminology would help interpretation. The same holds true for microcephaly which is a congenital abnormality.
- Page 6, line 26: how is co-infection a health outcome? Is the consequence of co-infection a health outcome and if so, is there any information available?
- The sentence page 7, line 10: "MC cases per birth, live births and prevalence ..." is difficult to understand, consider rephrasing.
- Page 7, line 11: "MC risk". Please be more precise. MC risk after maternal (suspected) ZIKV infection? (any trimester?).
- Page 7, line 26: the authors mention 5 SRs, but cite only 2.
- Page 7, line 45: the incidence cited from Simoes et al. seems much lower and close to an expected baseline incidence of GBS in absence of ZIKV outbreak. How should we interpret this?
- Page 7, line 50: incidence without denominator.
- The author seem to report percentages and proportions, would it make sense to go with one form?
- For some outcomes the authors mention sample sizes, for others they don't. It would help to have number of studies, study design and sample sizes to illustrate the robustness of the numbers produced.
- Paragraph on 'neurological complications': this paragraph does not open with a description of the number of SRs reporting on these outcome?
- Do other auto-immune outcomes, such as ITP have a place in the results?

Discussion

The discussion is very limited as it is, not putting results in context, nor citing comparative work.

- What overlap is there in the evidence considered? Does summarizing the evidence inflate the results of studies that were included multiple times?
- What type of bias do you expect to hamper the conclusions?
- Could the authors provide recommendations on preventing the most common?
- What do others say on the quality of SRs in similar context?
- Agreement between the studies?
- Can the studies be compared, since the objectives might vary?
- Do you believe publication bias would result in just a listing of most common outcomes, ignoring rather rare outcomes?

Judgement of the novelty of the findings presented in this paper: Do the results bring us anything new regarding the health outcomes? No. We knew before this study that the most common health outcomes were the ones they list. However, we do need critical assessment of systematic reviews on ZIKV-related outcomes.

Do the results bring us anything new regarding the critical assessment of the existing SRs? Possibly. Although the

	manuscript currently lacks the information to judge the validity of the conclusions of the authors. A message that SRs can be improved and highlighting elements where this can be done could be a way forward. Specific comments to tables and figures Figure 1: Please check the PRISMA flowchart, removing 339 duplicates results in a higher number of citations? Please provide reasons for exclusion as described in the PRISMA guidance document. Is there a reason labels (identification, screening, eligibility and included) are omitted? Table 1: It would make sense to give an overview of the outcomes described in the studies, more than 'jurisdictions'. The order of the studies seems random, nor does the table provide numerical references as in the text. It would be elegant to match this. Since ZIKV research is evolving, the timespan of included studies can be considered relevant here as well, would the authors consider including the inclusion date of the review (up to which date evidence was considered)? Krauer et al. in the supplement clearly provide the location of the studies, can the authors justify 'not clearly' reported? Figure 2: Consider identifying domains or individual questions (see below). The order of the studies does not seem to match Table 1, or the supplementary information.
--	---

VERSION 1 – AUTHOR RESPONSE

Reviewer: 1

Reviewer Name: Lavinia Schuler-Faccini

Institution and Country: Universiade Federal do Rio Grande do Sul

Please state any competing interests or state 'None declared': None Declared

Please leave your comments for the authors below

This is an interesting manuscript that summarizes correctly the outcomes related to ZIKV infection, mostly after gestational exposures. Moreover, performing a systematic review of systematic reviews the authors were able to detect the weaknesses in other published papers with similar approach. Another strength is that they didn't had limits to English language; in this particular topic it's very helpful since there are publications in Portuguese and Spanish.

Thank you!

I have only one minor observation: p3; In 26-27 (ref 10) the authors mention that the ZIKV and associated microcephaly occurred in 2016 in South America. Actually it was 2015-2016. In Brazil the peak of microcephaly was in 2015 decreasing in 2016. Colombia and other countries followed the peak in 2016. Although the formal establishment of ZIKV as a potential teratogen was in 2016, it was largely based from observations of babies born in 2015.

Updated as requested.

Reviewer: 2

Reviewer Name: Michel Counotte

Institution and Country: Institute of Social and Preventive Medicine, University of Bern, Bern, Switzerland

Please state any competing interests or state 'None declared': None declared

Please leave your comments for the authors below

Ximenes et al. describe the health outcomes associated with Zika virus (ZIKV) infection in humans. They use a systematic review of systematic reviews (SR) to make a systematic inventory of the available published work indexed in the MEDLINE, Embase or Cochrane database up to February 13, 2019. They provide an overview of the different ZIKV-associated health outcomes from 15 SRs. The authors describe the results presented in individual SRs. They conclude that higher quality SRs are required, based on an evaluation of the quality using the AMSTAR2 tool.

The authors take on the enormous task of navigating through 15 SR reporting on 667 studies (likely with quite some overlap), which should be applauded. However, the description of the studies and discussion as it is now should be improved.

Thank you!

Introduction

- Page 3, line 38 on the purpose of the study: Do the authors mean a causal association between ZIKV and health outcomes? For the sake of public health guidance, the definition of the type of association might be relevant and at least worth to clarify.

Thank you. We were interested in outcomes associated with ZIKV infection more broadly given the emerging nature of the field. However, we do comment on causality in the discussion based on the reviewed literature: "The included SRs indicate that ZIKV infection is causally associated with congenital abnormalities, including microcephaly, and that ZIKV infection is a trigger of GBS, considering evidence on biological plausibility, the strength of association, and the exclusion of alternative explanations."

Methods

- We are currently almost half a year after the inclusion date, would the authors consider updating the review?
Would the authors consider searching "LILACS" as an additional literature database?

Updated as requested.

Results

- General comment: A more intuitive structure might be to present the results first and the risk of bias assessment last?

Updated as requested.

- One could argue that 'risk of bias assessment' is not the right way to describe the AMSTAR2 conclusions, since these partly check reporting quality, and partly bias.

Thank you. We agree and have changed the terminology to "quality assessment".

Overview of included studies:

- It seems that the SRs included are conducted with different research questions or PICOs guiding them. It would be great to provide an overview of these in the supplementary material.

Thank you. We have added this information in Table 1.

- Is there a reason for example Prata-Barbosa (2018) (<http://www.scielo.br/pdf/jped/v95s1/0021-7557-jped-95-s1-0s30.pdf>) was not included (review on the effect of Zika on growth)?

The authors specifically state that their search strategy was "non-systematic". For this reason, we did not include this study in our systematic review.

- It seems that on page 4, line 52 the authors distinguish between reported symptoms and outcomes, and use this as exclusion criterion, but continue to discuss reported symptoms as

a health outcome. This seems contradictory. Either revisit the inclusion criteria or omit discussing symptoms.

Thank you. We have decided to omit the reported symptoms.

Quality of included studies/AMSTAR rating

- Since the authors worked with independent reviewers: how was the agreement and disagreement between the reviewers in AMSTAR2 rating? Does this provide reasons for discussion? Can the authors discuss the performance of the tool?

Reviewer agreement was 91% for the quality appraisal. We have added this information in the manuscript. However, we did not formally assess the performance of AMSTAR 2 and therefore cannot comment on this.

- Can the authors provide insight in the AMSTAR2 rating, a more complete description, for example a supplementary table with ratings. The current figure only provides the percentage of scores within different categories. There is currently no way to see on which domains studies scored poorly or could be improved, this might also facilitate the discussion of the results, which is currently lacking.

We included a new table with the ratings for each study and the percentage of the 'yes', 'partial yes' or 'no' answers in the main text.

- It is unclear which items are the critical domains in figure 2. Please clarify. Page 5 line 23 only mentions 'the majority'. Since one of the main conclusions is on the quality of studies, this needs to be expanded. A table in the main text might be appropriate.

We updated Figure 2 and now show the results for all AMSTAR2 questions and the result for the critical domains only in two different panels.

- Does this quality rating satisfy all the needs? Are the authors interested in the health outcomes related to suspected ZIKV or confirmed ZIKV? Is this worth discussing as limitation that we are often not sure on the infection status?

Because of the emerging nature of ZIKV, we opted to include all studies, independent of the ZIKV case definition. Restricting to confirmed cases, while methodologically stronger, would omit potentially important information at this time. Future reviews may opt to only include studies with a clear case definition that requires included subjects to be confirmed cases. We agree with the reviewer that infection status is often unclear and have included this information in the discussion.

Overview of Health outcomes

General remarks:

- It seems that the authors mainly describe how many of the included SRs describe a certain health outcome, would the authors agree that a systematic description of the actual incidence or prevalence would be more informative?

Thank you. We generally agree; however, because of the heterogeneity of the included studies, synthesis is challenging. We have added ranges for outcome measures as much as possible throughout the results section. The supplementary file 2 contains all incidence/prevalence values obtained in this SR.

- We numbers are provided, denominators are often unclear: the authors report a 'prevalence' of an outcome, but fail to mention the denominator, making it difficult to interpret, consistency and clarification throughout the results would be in place.

We report findings as reported in the included SR. Sometimes, denominators are not reported in the reviewed SRs. In these cases, we now specify the lack of information in the text.

- A structured description of the results per outcome in the form of a table might clarify the reported estimates, and heterogeneity. The text seems to be a somewhat random inclusion of numbers that are rather hard to follow.

Thank you. We edited the table for clarity.

- The definition of 'congenital abnormalities' is confusing and seems to include microcephaly in some cases, but not in others?

We updated the table to improve clarity.

Specific remarks:

- Page 5, line 36-37 malformations or congenital abnormalities and brain abnormalities: the latter could be included in the first? Again: clarifying terminology would help interpretation. The same holds true for microcephaly which is a congenital abnormality.

Thank you. We edited the table for clarity.

- Page 6, line 26: how is co-infection a health outcome? Is the consequence of co-infection a health outcome and if so, is there any information available?

Thank you. We agree, co-infection is not a health outcome and we therefore moved this to a new section.

- The sentence page 7, line 10: "MC cases per birth, live births and prevalence ..." is difficult to understand, consider rephrasing.

Updated as requested.

- Page 7, line 11: "MC risk". Please be more precise. MC risk after maternal (suspected) ZIKV infection? (any trimester?).

Updated as requested.

- Page 7, line 26: the authors mention 5 SRs, but cite only 2.

Updated as requested.

- Page 7, line 45: the incidence cited from Simoes et al. seems much lower and close to an expected baseline incidence of GBS in absence of ZIKV outbreak. How should we interpret this?

Thank you for identifying a mistake. There was one case report in French Polynesia in which GBS was diagnosed in a patient with ZIKV, therefore, there is no information about the total number of patients. We have corrected this in the text.

- Page 7, line 50: incidence without denominator.

Unfortunately, the original paper does not report the denominator.

- The author seem to report percentages and proportions, would it make sense to go with one form?

Updated as requested.

- For some outcomes the authors mention sample sizes, for others they don't. It would help to have number of studies, study design and sample sizes to illustrate the robustness of the numbers produced.

Thank you. We added sample sizes where possible. Not all studies reported the denominator. Further, because of the heterogeneity across the included SRs we often report ranges where at times it may be difficult to also report supporting information in the text. However, all data is reported in Table 1.

- Paragraph on 'neurological complications': this paragraph does not open with a description of the number of SRs reporting on these outcome?

Updated as requested.

- Do other auto-immune outcomes, such as ITP have a place in the results?

Updated as requested.

Discussion

The discussion is very limited as it is, not putting results in context, nor citing comparative work.

- What overlap is there in the evidence considered? Does summarizing the evidence inflate the results of studies that were included multiple times?

Updated as requested.

- What type of bias do you expect to hamper the conclusions?

Thank you. We included a discussion on potential bias, especially as it relates to reporting less common outcomes.

- Could the authors provide recommendations on preventing the most common?

It is unclear what the reviewer is referring to.

- What do others say on the quality of SRs in similar context?

We did not identify other SRs of SRs for ZIKV-associated health outcomes and have not formally searched for and reviewed SRs of SRs for other infectious diseases and can therefore not comment.

- Agreement between the studies?

Updated as requested.

- Can the studies be compared, since the objectives might vary?

Updated as requested. We added to the discussion on the heterogeneity across SRs, including different objectives across the included SRs, especially in the context of an emerging disease.

- Do you believe publication bias would result in just a listing of most common outcomes, ignoring rather rare outcomes?

Thank you, we agree and updated the discussion.

Judgement of the novelty of the findings presented in this paper:

Do the results bring us anything new regarding the health outcomes? No. We knew before this study that the most common health outcomes were the ones they list. However, we do need critical assessment of systematic reviews on ZIKV-related outcomes.

Do the results bring us anything new regarding the critical assessment of the existing SRs? Possibly. Although the manuscript currently lacks the information to judge the validity of the conclusions of the authors. A message that SRs can be improved and highlighting elements where this can be done could be a way forward.

Specific comments to tables and figures

- Figure 1: Please check the PRISMA flowchart, removing 339 duplicates results in a higher number of citations? Please provide reasons for exclusion as described in the PRISMA guidance document. Is there a reason labels (identification, screening, eligibility and included) are omitted?

Thank you for spotting this mistake. We have corrected the PRISMA flow chart.

- Table 1: It would make sense to give an overview of the outcomes described in the studies, more than 'jurisdictions'. The order of the studies seems random, nor does the table provide numerical references as in the text. It would be elegant to match this.

Updated as requested.

- Since ZIKV research is evolving, the timespan of included studies can be considered relevant here as well, would the authors consider including the inclusion date of the review (up to which date evidence was considered)?

Thank you, we have included this information in Table 1.

- Krauer et al. in the supplement clearly provide the location of the studies, can the authors justify 'not clearly' reported?

Thank you. The location of studies is not always clear and may only be reported for a subset of studies included in a SR. We have updated Table 1 and provide more detailed information.

- Figure 2: Consider identifying domains or individual questions (see below). The order of the studies does not seem to match Table 1, or the supplementary information.

Updated.

VERSION 2 – REVIEW

REVIEWER	Michel Counotte Institute of Social and Preventive Medicine, University of Bern
REVIEW RETURNED	10-Sep-2019

GENERAL COMMENTS	Dear, Ximenes et al. have improved their manuscript. Most issues have been addressed satisfactorily. It seems that a new paragraph on the overlap of citations is added. This assesses overlap in any citation between the systematic review, and is not limited to the results/included studies of the system reviews. Thus, we don't know if these citations are used in the introduction (for example Duffy et al in reference 19 just occurs in the introduction). The purpose of this section might be improved by simply looking at the citations that were a yield of the review, and not part of the introduction/discussion, or highlight this as limitation. The newly added sentence in the discussion that the low quality of SR may indicate publication bias might require some explanation.
---

VERSION 2 – AUTHOR RESPONSE

Reviewer: 2

Reviewer Name: Michel Counotte

Institution and Country: Institute of Social and Preventive Medicine, University of Bern, Bern, Switzerland

Please state any competing interests or state 'None declared': None declared

Please leave your comments for the authors below

Dear,
Ximenes et al. have improved their manuscript. Most issues have been addressed satisfactorily.

It seems that a new paragraph on the overlap of citations is added. This assesses overlap in any citation between the systematic review, and is not limited to the results/included studies of the system reviews. Thus, we don't know if these citations are used in the introduction (for example Duffy et al in reference 19 just occurs in the introduction). The purpose of this section might be improved by simply looking at the citations that were a yield of the review, and not part of the introduction/discussion, or highlight this as limitation.

Thank you.

All the studies included in the overlap analysis are included/result studies of the systematic reviews. The cited study (Duffy et al.) appear in the reference 19 in section 2 besides the introduction, where the study answers the questions: "2. What is the association between Zika virus and Guillain-Barré syndrome? Is it different during pregnancy? Can it affect the fetus?".

Taking the comment into consideration, we updated the paragraph for a better reader understanding.

The newly added sentence in the discussion that the low quality of SR may indicate publication bias might require some explanation.

VERSION 3 – REVIEW

REVIEWER	Michel Counotte Institute of Social and Preventive Medicine (ISPM)
REVIEW RETURNED	23-Sep-2019
GENERAL COMMENTS	The point last mentioned of the overlapping citations that only appear in introduction/discussion of SRs and not in their included studies remains open. However, this does not affect the message of the paper.

VERSION 3 – AUTHOR RESPONSE

Reviewer: 2

Reviewer Name: Michel Counotte

Institution and Country: Institute of Social and Preventive Medicine, University of Bern, Bern, Switzerland

Please state any competing interests or state 'None declared': None declared

Please leave your comments for the authors below

The point last mentioned of the overlapping citations that only appear in introduction/discussion of SRs and not in their included studies remains open. However, this does not affect the message of the paper.

Dear,

Ximenes et al. have improved their manuscript. Most issues have been addressed satisfactorily.

It seems that a new paragraph on the overlap of citations is added. This assesses overlap in any citation between the systematic review, and is not limited to the results/included studies of the system reviews. Thus, we don't know if these citations are used in the introduction (for example Duffy et al in

reference 19 just occurs in the introduction). The purpose of this section might be improved by simply looking at the citations that were a yield of the review, and not part of the introduction/discussion, or highlight this as limitation.

Thank you.

All the studies included in the overlap analysis are included/result studies of the systematic reviews. The cited study (Duffy et al.) appear in the reference 19 in section 2 besides the introduction, where the study answers the questions: “2. What is the association between Zika virus and Guillain-Barré syndrome? Is it different during pregnancy? Can it affect the fetus?”.

Taking the comment into consideration, we updated the paragraph for a better reader understanding.